# Tissue-Specific Metabolic Reprogramming during Wound-Induced Organ Formation in Tomato Hypocotyl Explants

**DOI:** 10.3390/ijms221810112

**Published:** 2021-09-18

**Authors:** Eduardo Larriba, Ana Belén Sánchez-García, Cristina Martínez-Andújar, Alfonso Albacete, José Manuel Pérez-Pérez

**Affiliations:** 1Instituto de Bioingeniería, Universidad Miguel Hernández, 03202 Elche, Spain; elarriba@umh.es (E.L.); ana.sanchezg@umh.es (A.B.S.-G.); 2Department of Plant Nutrition, Centro de Edafología y Biología Aplicada del Segura (CEBAS-CSIC), Campus Universitario de Espinardo, Espinardo, 30100 Murcia, Spain; cmandujar@cebas.csic.es (C.M.-A.); alfonsoa.albacete@carm.es (A.A.)

**Keywords:** de novo shoot apical meristem formation, de novo root regeneration, glycolysis/gluconeogenesis, photorespiration, photosynthesis, time-course bulk RNA-Seq

## Abstract

Plants have remarkable regenerative capacity, which allows them to survive tissue damage after exposure to biotic and abiotic stresses. Some of the key transcription factors and hormone crosstalk mechanisms involved in wound-induced organ regeneration have been extensively studied in the model plant *Arabidopsis thaliana*. However, little is known about the role of metabolism in wound-induced organ formation. Here, we performed detailed transcriptome analysis and used a targeted metabolomics approach to study de novo organ formation in tomato hypocotyl explants and found tissue-specific metabolic differences and divergent developmental pathways. Our results indicate that successful regeneration in the apical region of the hypocotyl depends on a specific metabolic switch involving the upregulation of photorespiratory pathway components and the differential regulation of photosynthesis-related gene expression and gluconeogenesis pathway activation. These findings provide a useful resource for further investigation of the molecular mechanisms involved in wound-induced organ formation in crop species such as tomato.

## 1. Introduction

Plants have remarkable regenerative capacity, enabling them to either repair damaged tissues or regenerate lost organs [1]. This regenerative capacity allows plants to overcome tissue damage caused by different environmental insults. Moreover, enhancing plant regeneration capabilities has been frequently used in crop production to increase yields by grafting and pruning without killing the plant, as well as for the clonal propagation of elite genotypes.

The many advantages of *Arabidopsis thaliana* and the wide repertoire of regenerative responses in this species have provided some insight into the molecular mechanisms underlying tissue repair and organ regeneration [2,3]. Two broad categories of regeneration models have been established; hereafter, these models are referred to as tissue culture-induced regeneration and wound-induced regeneration [4]. Key transcription factor families and hormone crosstalk mechanisms involved in plant regeneration have been extensively studied in *A. thaliana* [5]. In metazoans, the transcription factor-mediated reprogramming of somatic cells into induced pluripotent stem cells [6] involves a profound metabolic shift from oxidative phosphorylation to glycolytic-dependent energy generation [7,8]. In addition, the rewiring of energetic metabolism in cancer cells has been proven essential for tumor progression [9]. Strikingly, in plant tissue culture-induced regeneration, the addition of sugars, amino acids, and polyamines to the growth medium often influences cellular proliferation and regeneration efficiency. Indeed, sucrose is frequently chosen in cell and tissue culture media with positive effects on callus proliferation and shoot regeneration, although the precise mechanisms underlying these effects are currently unknown [10]. Ectopic expression of *A. thaliana WOUND INDUCED DEDIFFERENTIATION1* (*WIND1*) in hypocotyl explants of *Brassica napus* enhanced callus formation and root regeneration, and this phenotype correlated with the deregulation of metabolites, such as sucrose, proline, gamma aminobutyric acid, or putrescine [11]. However, the precise link between the activation of a single transcription factor, accompanied by non-hormone metabolite accumulation and callus formation and regeneration, requires additional investigation.

Tomato (*Solanum lycopersicum* L.) ‘Micro-Tom’ is an established model for tomato research because it shares some key advantages with *A. thaliana*; it is small, has a short life cycle, and is easy to transform [12]. High-efficiency targeted gene editing using an optimized CRISPR/Cas9 system [13] and TILLING collections [14] facilitates the use of reverse genetics approaches. In our previous studies on wound-induced adventitious root (AR) formation in tomato [15,16], we observed the formation of de novo shoot meristems at the cut surface of the hypocotyl after the removal of the shoot apex. Here, we performed detailed transcriptome analyses and used a targeted metabolomics approach to study de novo organ formation (i.e., adventitious shoot and AR formation) in tomato hypocotyl explants to provide a conceptual outline for the identification of the regulatory mechanisms involved in organ regeneration. The findings of this study reveal the role of metabolism regulation in tissue-specific wound-induced organ formation in tomato.

## 2. Results

### 2.1. Time-Course RNA-Seq Analysis of Wound-Induced Organ Formation in Tomato Hypocotyl Explants

We previously studied wound-induced AR development in the basal region of ‘Micro-Tom’ shoot explants [16]. We observed formative divisions of the cambial cells facing the phloem 1 day after wounding. AR primordia emerged from specific domains within the basal region of the explants after 3–4 days, and 96.8% of the explants developed between two and four ARs at 8 days (Figure 1a). Interestingly, hypocotyl explants obtained by sectioning shoot explants just below the cotyledons developed new functional shoots at their apical end after 3 weeks (Figure 1b). We confirmed the absence of preformed shoot apical meristems (SAMs) in these explants, which resulted in rapid sealing of the wound in the apical region (Figure 1c, stage 1). We observed that homogeneous callus-like tissue formation occurred beginning on 4–5 days and continuing thereafter (Figure 1c, stage 2). After 8 days, some regionalization within these calluses occurred, and bud outgrowth was obvious (Figure 1c, stage 3), which later resulted in new shoot apices (Figure 1c, stage 4).

To determine the molecular signatures during de novo organ formation in ‘Micro-Tom’, we performed a time-course bulk RNA-Seq bioinformatics analysis of the apical and basal regions of the hypocotyl explants at 0, 1, 4, and 8 days after excision (dae; Appendix A). Bayesian estimation of temporal regulation (BETR) with fold discovery rate (FDR) = 0.05 revealed that 20,831 genes, representing 60% of the tomato genes in the ITAG4.0 annotation, exhibited significant transcriptional changes. Approximately 10% of the transcripts were long noncoding RNAs (lncRNAs; Appendix A). The results of read count normalization statistical analysis are shown in Appendix A. We validated the RNA-Seq gene expression results from the apical and basal regions by RT-qPCR (Appendix A), and these results supported the reliability of our RNA-Seq dataset. Principal component (PC) analysis of the expression profiles revealed differences in transcript profiling both between the apical and basal regions and throughout the time course (Figure 1d).

### 2.2. Functional Enrichment Analysis of Gene Expression Profile during Wound-Induced Regeneration

To characterize the gene expression profiles associated with each regeneration stage, we carried out k-means clustering analysis using the most variable expressed genes (standard derivation > 0.5); this analysis included 4500 and 4600 genes from the apical and basal regions, respectively (Appendix A). The optimal cluster number was estimated using the Elbow method, which was conducted with iDEP.91 [17] (Appendix A). Three of these clusters displayed the same temporal profile in the apical and basal regions, corresponding to genes mostly expressed at T0 (cluster 1), genes expressed at T1 (cluster 2), and genes expressed at T4 and T8 (cluster 3; Figure 1e,f). In addition, we found two additional clusters in the apical region, which included genes expressed at T0–T1 (cluster 4) and T1–T4–T8 (cluster 5; Figure 1e). Venn diagrams and Gene Ontology (GO) enrichment analysis of clusters 1–3 allowed us to identify either tissue-specific (apical or basal) or shared GO enrichment subsets (Appendix A).

In this regard, the genes in cluster 1 that were shared between the apical and basal regions (10.6%; Appendix A) were related to auxin signaling and gibberellin biosynthesis, which are known to contribute to hypocotyl growth [18]. On the other hand, we identified some GO categories that were specifically enriched in the apical or basal regions, which indicate tissue-specific differences at T0. Cluster 2 might contain key regulators of de novo organ formation, as the expression levels of these genes transiently increased 24 h after wounding (Figure 1e,f). Despite the higher number of shared genes in cluster 2 (14.7%; Appendix A), we did not observe significant GO enrichment of these genes (Appendix A), which was consistent with the results of our PC analyses (Figure 1d) and with the divergent developmental patterns observed in the apical and basal regions after wounding. Regarding genes in cluster 2 that were specific to the apical region, we observed GO enrichment in the light signaling pathway and chromatin remodeling networks (Figure 2a and Appendix A). On the other hand, GO enrichment of terms related to oxidation–reduction processes, hormone biosynthesis, and hormone regulation was observed for genes specific to the basal region (Appendix A). We identified a functional GO enriched network related to cell-cycle regulation (Figure 2b and Appendix A) in the shared genes in cluster 3 (17.9%; Appendix A). Nevertheless, genes specific to the apical region were enriched in GO terms related to photosynthesis and de novo protein folding (Figure 2c and Appendix A). On the other hand, highly enriched GO terms related to cell-wall modification and reactive oxygen species (ROS) detoxification (Appendix A) were observed for genes specific to the basal region. Cluster 4 genes were mostly expressed in the apical region up to 24 h after excision (T1) and were enriched in terms related to cell-wall and polysaccharide metabolism (Figure 1e and Appendix A). Cluster 5 genes, which were highly expressed in the apical region between T1 and T8, were enriched in GO terms related to carbon metabolism-related biosynthetic processes (Figure 1e and Appendix A).

Our GO-enrichment analysis based on k-means gene clustering suggests that both shared and specific biological processes take place in the apical and basal regions of explants during de novo organ formation.

### 2.3. Differentially Expressed Genes during Wound-Induced Regeneration

In addition to k-means analysis, we identified 12,406 differentially expressed genes (DEGs) in different contrasts of samples collected at T0 (Figure 2d and Appendix A); these DEGs represented 59.6% of the expressed genes in our RNA-seq results. In the apical region of the explants, we found 8792 DEGs (4658 and 4265 upregulated and downregulated genes, respectively; Figure 2e). Among the upregulated genes in the apical region from the different contrasts (Figure 2e), we observed the enrichment of GO terms associated with serine/glycine metabolism, sterol biosynthesis, and response to ROS (Appendix A). Interestingly, the downregulated genes in the T1–T0 contrast were enriched in GO terms related to photosynthesis (light-dependent reactions), while a subset of auxin response genes was upregulated in this contrast (Appendix A). Moreover, we found upregulated genes that were enriched in GO terms related to cell cycle regulation, some of which were shared between T1–T0 and T8–T0 contrasts (Appendix A). The downregulated genes shared between the T4–T0 and T8–T0 contrasts were enriched in GO terms related to cell-wall biogenesis and cell redox homeostasis (Appendix A). Lastly, consistent with leaf primordia formation in the apical region during stage 4, we observed GO enrichment in terms associated with leaf development for the upregulated genes in the T8–T0 contrast (Appendix A).

Regarding the basal region, we identified 9085 DEGs, including 4076 and 5233 upregulated and downregulated genes, respectively (Figure 2e). Within the upregulated genes shared among all the contrasts (Figure 2f), we observed an enrichment of GO terms associated with ion transport. Some GO terms related to photosynthesis were enriched in the downregulated DEGs in the T1–T0 contrast (Appendix A). GO terms related to protein translation, rRNA modification, and RNA processing were also enriched in the T1–T0 contrast (Appendix A). Based in our enrichment analyses, we found that a subset of DEGs involved in cell division was downregulated in the T1–T0 contrast, while other genes were upregulated in the T4–T0 contrast (Appendix A). We also observed differential regulation of genes assigned to GO terms related to the response to oxidative stress. Indeed, some genes involved in ROS detoxification (oxidoreduction process) were upregulated in the T4–T0 and T8–T0 contrasts, while genes involved in H_2_O_2_ responses were downregulated in all the contrasts (Appendix A). GO term enrichment of upregulated genes in the T4–T0 and T8–T0 contrasts indicated the involvement of these genes in cell-wall biogenesis, which follows the formation and emergence of AR primordia in the basal region.

Our functional analysis based on GO term enrichment from k-means clustering and DEGs revealed the contribution of functionally related pathways, such as photosynthesis, ROS homeostasis, and carbon metabolism [19,20].

### 2.4. Tissue-Specific Regulation of Photosynthesis during Wound-Induced Organ Formation

Our functional GO analysis revealed an enrichment in genes related to photosynthesis during de novo organ formation (Appendix A). In this regard, most explants grown in vitro develop low photosynthetic efficiency due to chlorophyll degradation and the presence of sugars in the medium. Due to the relevance of this metabolic process, we identified 90 genes in the photosynthesis-related KEGG pathway (Appendix A), and the expression of 76 of these genes (84.4%) was deregulated (Appendix A). Hierarchical clustering analysis of the gene expression values of photosynthesis-related genes (Figure 3a,b) revealed two predominant profiles in the apical region: genes that are mainly repressed at T1 but regain their expression at T4 and T8 (red cluster, Figure 3a,c), and genes whose expression is enhanced from T1 to T8 (purple cluster, Figure 3a,c). On the other hand, the expression of most photosynthesis-related genes in the basal region was strongly inhibited at T1 and mildly decreased at T4 and T8 compared with their expression at T0 (green cluster, Figure 3b,c). These results indicated differential regulation of genes related to the photosynthetic machinery in the apical and basal regions after wounding (Figure 3d–g and Appendix A). Our DEG analysis identified genes involved in regulating the plastoquinone pool (Appendix A), such as those encoding the NADPH subunits PNSB1 and PNSB4, or those encoding ferredoxins (FD) and the NADP^+^ reductases LFNR1 and PGR5-like, whose expression was upregulated in the apical region beginning at T1 (Figure 3d). In the apical region, the expression of most genes encoding the core components of PSII was upregulated from T1 onward (Figure 3e). However, the expression of most genes encoding different PSI proteins was downregulated in this region at T1 and was highly upregulated in the apical region at T4 and T8 (Figure 3f). Interestingly, the expression of genes encoding antenna proteins of both photosystems (LHCB and LHCA) was severely downregulated in the apical region only at T1 and was downregulated in the basal region beginning at T0 (Appendix A). In addition to deregulation of the genes associated with different components of the photosynthetic machinery in the apical region, the expression of regulatory genes encoding the protein kinases ABC1K1 and STN8, as well as the auxiliary PSII core protein PSB33, was also upregulated (Figure 3g). Conversely, in the basal region, the levels of all these genes were significantly downregulated beginning at T1 (Figure 3g), suggesting that photosynthesis function might be reduced in the basal region after wounding and during AR formation.

To determine the role of photosynthesis in wound-induced organ formation, we grew hypocotyl explants under different light and photoperiod conditions (Figure 3h). Hypocotyl explants grown in continuous darkness (0:24 photoperiod) were able to produce fewer ARs in the basal region of the hypocotyl, with a severe delay in their emergence (Figure 3i,j), but de novo organ formation in the apical region of these explants was retained at stage 2 (Figure 3h). We observed that hypocotyl explants incubated for 3 days under standard photoperiod conditions (16 h of light and 8 h of dark; 16:8) and then transferred to conditions of continuous darkness were able to overcome stage 2 and produce new shoots (Figure 3h). In addition, AR emergence and root capacity were both rescued by light (Figure 3i,j). Taken together, these results indicated that the photosynthetic machinery was differentially regulated in the apical and basal regions of hypocotyls after wounding and mainly contributed to de novo shoot formation in response to light.

### 2.5. Photorespiration Is Required for Wound-Induced Shoot Regeneration

We observed significant enrichment in GO terms associated with carbon metabolism (Appendix A). We found 315 expressed genes related to carbon metabolism, of which 199 were DEGs (Appendix A). Key metabolic pathways, such as glycolysis/gluconeogenesis, carbon fixation through the Calvin–Benson cycle, glycolate/glyoxylate metabolism (i.e., photorespiration), and pentose phosphate pathway, were deregulated in our bulk time-course RNA-Seq results (Appendix A). A subset of these DEGs showed contrasting deregulation between the apical and basal regions and were further studied. Because of the massive downregulation of components of the photosynthesis machinery at T1 (see above), the energy supply required for new organ growth might be compromised in the hypocotyl explants. Deregulation of genes encoding key enzymes of the Calvin–Benson cycle for CO_2_ fixation via 3-phosphoglycerate (3-PGA) showed opposite trends in the apical and basal regions, which suggests de novo establishment of sink–source relationships after wounding in these two regions (Appendix A).

Photorespiratory metabolism involves the production of 3-PGA by the oxygenase activity of the RuBisCO enzyme through the glycine and serine pathways, and this process occurs at a high metabolic cost [21]. Because most genes encoding RuBisCO subunits are highly expressed (Figure 4a) and atmospheric O_2_ floods the wounded tissue in the apical region, photorespiration should be active in this tissue. We found that the expression of several genes encoding key enzymes of the glycine biosynthesis pathway downstream of 2-phosphoglycolate, such as PGLP1, GLO4, and GGAT, was specifically upregulated in the apical region after wounding (Figure 4a). In contrast, the levels of all these genes were downregulated in the basal region (Figure 4a). Because of this differential regulation, glycine levels, as well as the levels of H_2_O_2_ as a subproduct, might substantially increase in the apical region. In the next step of the photorespiratory pathway, serine is produced via the glycine decarboxylase complex (GDC) and serine hydroxymethyltransferase (SHM) enzymes [21]. The expression of *GLP1/2*, which encodes the P-subunit of the GDC, and *SHM1* was upregulated in the apical region after wounding (Figure 4a). Serine is converted to hydroxypyruvate by the product of *AGT1*, and the expression of this gene was also upregulated in the apical region (Figure 4a). Finally, hydroxypyruvate is incorporated into the Calvin–Benson cycle as 3-PGA through the activities of the enzymes encoded by *HPR* and *GLYK* (Figure 4a). As was found for the previous steps, most of the other genes of the photorespiratory pathway were not deregulated in the basal region after wounding, which clearly indicates that photorespiration is limited to the apical region during wound-induced organ regeneration in tomato hypocotyl explants. To confirm that the observed changes in gene expression contribute to the spatial regulation of photorespiration, we measured the endogenous levels of three photorespiratory pathway intermediaries (glycolate, glyoxylate, and hydroxypyruvate) during de novo organ formation by targeted metabolome analysis (Appendix A). At wounding time (T0), we did not observe significant differences in the endogenous levels of these intermediaries between the apical and basal regions of the explants (Figure 4b). However, in all cases, we observed highly significant differences in the endogenous levels of these intermediaries throughout the time course, and the highest levels were observed in the apical region between T1 and T8 (Figure 4b). Consistent with these results, the 3-PGA levels were significantly enhanced in the apical region of the explants at T1 and T4, while there were no significant differences between the apical and basal regions at T8 (Figure 4c). 3-PGA is a key intermediary of both the Calvin–Benson cycle and glycolysis, and it is converted into glyceraldehyde 3-phosphate (G3P) by phosphoglycerate kinase (PGK) and glyceraldehyde-phosphate dehydrogenase (GAPA) (Figure 4a). We observed higher levels of 1,3-bisphosphoglycerate and glyceraldehyde 3-phosphate in the apical region of the explants (Figure 4c), which was consistent with the greater upregulation of *PGK* and *GAPA* gene expression in this region (Figure 4a). In addition, the genes encoding the enzymes involved in the subsequent regeneration of ribulose 1,5-bisphosphatase in the Calvin–Benson cycle were also differentially regulated in the apical and basal regions (Appendix A). To confirm the functional relevance of photorespiration during de novo shoot formation, we investigated the effect of local application of 20 mM 3-PGA or 10 mM isoniazide (INH), a known inhibitor of photorespiration [22]. Neither 3-PGA nor INH treatment affected AR emergence or rooting capacity at 14 dae (Figure 4d; *p*-value = 0.986). On the other hand, 3-PGA slightly enhanced de novo shoot formation, while INH treatment caused a significant delay in shoot regeneration (Figure 4e,f). Lastly, following an in silico approach (see Section 4), we identified 15 deregulated transcription factors in the apical region (Appendix A) whose binding sites were overrepresented in the promoters of genes of the photorespiration pathway and that were upregulated in the apical region (Figure 4a). Interestingly, some of these transcription factors (e.g., BZO2H3 and HY5) have been described in other species as being related to the regulation of light and energy crosstalk [23,24].

Taken together, these results indicate that the apical region undergoes strong metabolic reprogramming after wounding, with photorespiration being a specific pivotal metabolic pathway that is required for effective de novo shoot formation after wounding.

### 2.6. Spatial Regulation of Sugar Metabolism during Wound-Induced Organ Formation

Our data indicated that photorespiration sustains energy production in the apical region until the reactivation of photosynthesis, and we wondered whether the presence of sucrose in the medium also contributed to the observed regeneration response of the explants. Uptake of sugars is mediated by invertases, which irreversibly hydrolyze sucrose into glucose and fructose [25] and are classified according to their subcellular localization in the cell wall or as cytosolic and vacuolar invertases (CWIN, CIN, and VIN; Appendix A) [26,27]. In addition, sucrose cleavage is mediated by cytosolic sucrose synthase (SUS) [28]. The expression of most genes encoding CIN, VIN, and SUS enzymes was downregulated in the apical and basal regions after wounding (Figure 5a). On the other hand, the expression of several *CWIN* genes was upregulated in both regions (Figure 5a). Extracellular glucose and fructose are taken up by plant cells via hexose transporters [29,30]. We found that the expression of several genes encoding SWEET transporters (e.g., *SWEET10b*, *10c*, *11a*, *12a*, *12b*, and *12c*) [31] was upregulated in the apical and basal regions of the explants throughout the time course (Figure 5b and Appendix A). Additionally, the expression of *SUT1*, whose product is involved in apoplastic phloem loading [32], was specifically upregulated in the apical region over time (Figure 5b).

The metabolic activation of glucose and fructose involves the hexokinase (HK) and fructokinase (FK) enzymes, respectively, which were expressed and were differentially regulated in our RNA-seq results (Figure 5c and Appendix A). The first downstream regulatory step of the glycolysis pathway involves ATP-dependent phosphofructokinase (PFK) and pyrophosphate-dependent phosphofructokinase (PFP). We identified eight genes that encode PFK enzymes and five genes that encode PFP enzymes (Appendix A), most of which were upregulated in both regions throughout the time course (Figure 5c). The expression of several genes that encode pyruvate kinase (PK), another key regulatory enzyme of the glycolytic pathway, was similarly upregulated in the apical and basal regions after wounding (Figure 5c). These results suggest that sucrose is catabolized to pyruvate in both the apical and the basal regions, while photorespiration provides a surplus of G3P that might also be used for pyruvate production mostly in the apical region (see previous section). Indeed, the pyruvate levels slightly increased in the apical region but were not significantly affected in the basal region throughout the time course (Figure 5d). Importantly, gluconeogenesis might also occur from pyruvate or oxaloacetate, the latter being a direct product of the citrate cycle or the glycolate/glyoxylate cycle. Indeed, the expression of the gene encoding pyruvate, phosphate dikinase (PPDK), was upregulated in the apical region of the explants during over time (Figure 5c). Another key regulatory factor in gluconeogenesis is fructose 1,6-bisphosphatase (FBPase), and we found three FBPase-encoding genes whose expression was specifically upregulated in the apical region of the explants beginning at T1 (Figure 5c). Taken together, these results confirm the metabolic switch of the apical region from being a sink tissue to becoming a source tissue.

### 2.7. De Novo Organ Formation in Tomato Hypocotyl Explants Depends on Sugar Availability

Lastly, since we showed that both photosynthesis and sucrose metabolism are deregulated during wound-induced organ formation, we investigated the relevance of these two factors during de novo organ formation in tomato hypocotyl explants. Hypocotyl explants grown in the dark (0:24 photoperiod) on sucrose-depleted medium did not produce ARs, and de novo shoot formation in their apical region remained in stage 1 (Figure 5e). In explants grown under standard photoperiod conditions (16:8) on sucrose-depleted medium, ARs were produced in the basal region of half of the explants, but de novo shoot formation in the apical region was not observed (Figure 5e). To confirm that the observed effect of sucrose on wound-induced organ formation depends on the glycolysis pathway, we grew hypocotyl explants on media supplemented with different sugars for 10 days in the dark (Appendix A). When grown in the presence of sucrose, fructose, or glucose, ARs rapidly emerged, and, by 5 dae, all the explants had produced at least one AR (Figure 5f). Additionally, hypocotyl explants grown on sorbitol-supplemented medium did not produce any ARs until the explants were transferred to a 16:8 photoperiod at 11 dae (Figure 5f). At the end of the experiment (17 dae), we observed that hypocotyl explants were able to produce de novo shoots (stage 3) at similar proportions (31–50%) regardless of the sugar that was added (Figure 5g). We inhibited glycolysis by incubating the hypocotyl explants with 2-deoxyglucose (2-DG), and AR emergence was completely blocked (Figure 5h,i). In addition, de novo shoot formation was not observed in the apical region of these explants, which remained in stage 1 (Figure 5i). These results indicate that exogenous sucrose was essential for effective shoot regeneration in the apical region of the hypocotyl explants but was not necessary for AR formation when another energy source was present (e.g., photosynthesis/photorespiration-derived compounds, such as 3-PGA and G3P).

## 3. Discussion

We established here a new experimental system using ‘Micro-Tom’ hypocotyl explants to simultaneously study wound-induced de novo shoot formation and AR development without the exogenous application of plant hormones. We employed bulk time-course RNA-seq and target metabolite profiling to characterize tissue-specific reprogramming events in ‘Micro-Tom’ hypocotyl explants. Time-course transcriptome analyses of wound-induced de novo organ formation have been previously performed in *A. thaliana* [33,34,35,36]. In addition, transcriptome dynamics during tissue healing after wounding have been extensively studied in this model species [37,38,39]. These earlier studies have contributed to the understanding of some of the hormonal crosstalk that drives the different tissue reprogramming events, and they have allowed the identification of some of the transcriptional regulators involved in these processes, but comparative studies in other plant species are limited. Our transcriptome data indicate both tissue-specific differences and divergent metabolic patterns in the apical and basal regions of ‘Micro-Tom’ hypocotyl explants after wounding. We also observed significant variation in lncRNA expression across the studied tissue regions and over the time course, suggesting a regulatory role for these non-protein-coding genes in wound-induced organ formation; this role will be addressed in future studies. In our experimental system, AR formation in the basal region of hypocotyls relies on the activation of resident stem cells due to rapid auxin homeostasis regulation after wounding [16], while, in the apical region, callus formation and metabolic reprogramming precede de novo shoot formation (this work).

We observed differential regulation of photosynthesis-related genes in the apical and basal regions of ‘Micro-Tom’ hypocotyls after wounding. On the one hand, photosynthesis-related gene expression was strongly downregulated in the basal region over time. On the other hand, the expression of most of the photosynthesis-related genes was constitutively upregulated in the apical region after wounding, except those encoding antenna proteins (LHC) and PSI components, which were transiently downregulated at T1. As these conditions resembled adaptation to high-light stress [40], the high PSII/PSI ratio in the apical region of the hypocotyls shortly after wounding might lead to the formation of singlet oxygen and, thus, cause photooxidative damage. Consistent with the dynamic regulation of photosynthesis function in the apical region after wounding, the expression of key regulators of light acclimation (ABC1K1, STN8, and PSB33) was specifically upregulated in this tissue. Our results also suggest that light is a novel photo-regenerative factor. LONG HYPOCOTYL5 (HY5) participates in a systemic signaling mechanism that triggers photoprotection in fluctuating light environments [23], likely by regulating genes associated with PSII function [41] or by regulating the homeostasis of carbon and nitrogen metabolism [42]. Interestingly, our in silico analysis of the photorespiration pathway during wound-induced regeneration also suggested a regulatory function for HY5 in this process. DE-ETIOLATED1 (DET1) is a major repressor of photomorphogenesis that is associated with CULLIN4 (CUL4)-based E3 ubiquitin ligases for the degradation of several transcription factors [43]. Indeed, a novel regulatory network involving DET1–COP1–HY5 has been described in *A. thaliana* photomorphogenesis [44]. Whether a conserved regulatory module downstream of DET1 and involving HY5 is required for wound-induced de novo shoot formation in tomato hypocotyl explants needs to be addressed.

Reprogramming glucose metabolism in cancer cells, even in the presence of atmospheric oxygen (the Warburg effect), is a key event that is required for sustaining tumor growth, as high glycolytic flux provides sufficient energy and the metabolic intermediates required by rapidly proliferating cells [9,45,46]. In addition, cancer cells rely on the serine/glycine biosynthetic pathway to produce one-carbon (C1) metabolites that are required for tumorous growth [47,48]. We demonstrated that wound-induced organ formation in tomato hypocotyl explants, i.e., both ARs and de novo shoot formation, was dependent on the sugar supply. The expression of several sugar transporters of the SWEET family and CWIN was constitutively upregulated in both the apical and the basal regions of hypocotyls after wounding. As observed during heart regeneration in zebrafish [49], blocking glycolysis with 2-deoxyglucose impaired the ability of cells near the wound to produce new organs. These results suggest that glycolysis-derived energy (ATP) drives cell proliferation in tomato hypocotyl explants, which is required for wound-induced organ formation; thus, these tissues are rapidly reprogrammed into sink tissues.

We found specific upregulation of photorespiratory pathway components in the apical region of tomato hypocotyl explants after wounding and during de novo shoot formation. Photorespiration recycles toxic 2-phosphoglycolate into 3-PGA, which can enter the Calvin–Benson cycle and is a key metabolite that drives C1 metabolism in plants [50]. We hypothesized that initial callus growth in the apical region of the hypocotyl relies on photorespiration-produced 3-PGA, which could also be used as a substrate for gluconeogenesis. Indeed, we observed differential upregulation of the expression of key genes related to gluconeogenesis (*PPDK*, *FBPase*), which correlated with increased levels of the intermediates of these two pathways (glyoxylate, hydroxypyruvate, 3-PGA, G3P) in the apical region of the explants. Through the local inhibition of photorespiration in the apical region, we provided additional evidence of a functional role of this pathway in sustaining proliferation and callus formation, which preceded and was required for wound-induced shoot initiation. However, we could not exclude the possibility that photorespiration might alter redox homeostasis [51], which might directly affect de novo shoot formation. Furthermore, photorespiration can counteract the negative effect of the transient downregulation of photosynthesis-related genes in the apical region, consistent with the alleged beneficial role of photorespiration in dissipating excessive energy under abiotic stress conditions [52]. In this regard, the BZO2H3 transcription factor identified in our in silico approach has been recently associated with the regulation of transcriptional changes induced by energy deprivation in *A. thaliana* [24], which mediates the metabolic adjustment of the circadian oscillator [53]. Our results highlight a central role of photorespiration in wound-induced de novo shoot formation, and additional experiments are required to assess its role as ROS producer, photosynthetic machinery protector, or alternative energy source.

## 4. Materials and Methods

### 4.1. Plant Material and Growth Conditions

Seedlings of the tomato ‘Micro-Tom’ were grown in vitro as described elsewhere [16]. Briefly, seeds were surface sterilized in 2% (*w*/*v*) NaClO. Seeds were then transferred to wet chambers at 28 °C in a dark growth cabinet for 96 h. Germinated seedlings with primary cotyledons and roots were transferred to 65 × 120 mm (diameter × height) glass jars containing 75 mL of sterile one-half-Murashige and Skoog basal salt medium (SGM, Appendix A), which were transferred to a growth cabinet with 16 h light (average photosynthetic photon flux density of 50 μmol·m^−2^·s^−1^) at 26 ± 1 °C and 8 h darkness at 23 ± 1 °C. Hypocotyl explants were obtained after removing the whole root system (2–3 mm above the hypocotyl–root junction) and the shoot apex (just below the petioles of the cotyledons) with a sharp scalpel at the 100–101 growth stages (9–11 day old seedlings) [54]; hereafter, this timepoint is referred to as 0 days after excision (dae). The hypocotyl explants were transferred to 120 × 120 mm (length × width) Petri dishes containing 75 mL of standard growth medium (SGM) [16], unless otherwise indicated (Appendix A).

### 4.2. RNA Isolation, Library Construction, and NGS Sequencing

For each sample, 3–4 mm of the apical or basal regions of the hypocotyl were collected at 0, 1, 4, and 8 dae (denoted as T0, T1, T4, and T8). Biological replicates consisting of 12 apical or basal fragments were harvested and immediately frozen in liquid N_2_. Total RNA was extracted from ~100 mg of powdered tissue using the Spectrum™ Plant Total RNA Kit (Merck, Burlington, MA, USA) and treated with DNAse I (Thermo Fisher Scientific, Waltham, MA, USA); the RNA was then stored at −80 °C. The RNA integrity was confirmed using a 2100 Bioanalyzer (Agilent Technologies, Santa Clara, CA, USA). Three sequencing libraries from basal tissues and two sequencing libraries from apical tissues at T0, T1, T4, and T8 were constructed with the TruSeq Stranded RNA Sample Preparation Kit v2 (Illumina, San Diego, CA, USA). Next-generation sequencing (NGS) was carried out by Macrogen (Seoul, Republic of Korea) on an Illumina HiSeq4000 in paired-end mode with 100 cycles of sequencing. Additionally, three NGS libraries were sequenced using the BGISEQ-500 pipeline (BGI-Tech, Shenzhen, China), in pair-end mode with 150 sequencing cycles. The raw NGS reads were preprocessed using Trimmomatic [55], and FastQC [56] was used for quality assessment. We sequenced three libraries for the basal region and two libraries for the apical region (BioProject: PRJNA731333). Each library was constructed with 12 different individuals obtained from three biological replicates (i.e., three independent regeneration experiments performed in the lab).

### 4.3. RNA-Seq Analysis

The bioinformatics workflow used in this study is shown in Appendix A. Briefly, clean RNA-Seq reads were mapped to the *S. lycopersicum* genome build SL4.0 [57] using STAR 2.7 [58]. Assignation of the reads to gene models (ITAG4.0 annotation) was performed with featureCounts from the Subread package [59]. Identification of significant transcriptional changes in gene expression was carried out on the basis of BETR analysis with a false discovery rate (FDR) = 0.05 [60] in the MeV software package [61]. Read count normalization and differential gene expression analysis were carried out using DESeq2 integrated into the Differential Expression and Pathway analysis (iDEP 9.1) web application [17]. Statistical analysis and read count normalization are shown in Appendix A. Differentially expressed genes (DEGs) were filtered using an FDR < 0.01 and log_2_ fold change > |1| for T0 (Appendix A). K-means clustering and PC analysis were carried out using normalized counts on the iDEP 9.1 web application. The web-tool ShinyGO v0.61 [62] was used to perform GO enrichment analysis. ITAG4.0 gene structural annotation was retrieved from the Sol Genomics Network [63]. Their putative *A. thaliana* orthologs were identified using the BioMart tool from the Emsembl Plants database [64]. Genes were assigned to KEGG metabolic pathways by reciprocal best hit BLAST using GhostKOALA [65] and SolGenomics BLAST against ITAG4.0. For heatmap and hierarchical clustering analyses, we used Morpheus [66]. The identification of over-represented transcription factor binding sites shared by sets of co-expressed genes (*p*-value < 0.05) was carried out on 2 kb promoter sequences upstream of the transcription start site identified in the SL4.0 assembly using the Regulation Prediction web tool from PlantRegMap [67].

### 4.4. Gene Expression Analysis by Real-Time Quantitative PCR

Total RNA was extracted in triplicate from ~100 mg of powdered hypocotyl sections from 14 seedlings at T0, T1, T4, and T8 using the Spectrum Plant Total RNA Kit (Merck) and further processed as described elsewhere [16]. For real-time quantitative PCR (RT-qPCR), primers were used to amplify 115–205 base pairs of the cDNA sequences of the selected genes (Appendix A). RT-qPCR was performed as described previously [16].

### 4.5. Macroscopic Studies of Wound-Induced Organogenesis

Hypocotyl explants were incubated in SGM supplemented with different compounds for 10 to 21 days (Appendix A). In all these cases, ARs arising from the hypocotyl were visually scored and periodically annotated. AR emergence was estimated on the basis of the day of the observation of the first AR. Shoot regeneration stages were also scored according to the morphological structures observed in the apical region of the hypocotyl. For photorespiration assays, we applied the treatments in 0.2% agarose droplets (25 µL) at the apical region of the hypocotyl explants; the shoot regeneration stages were periodically scored until 17 dae.

### 4.6. Metabolite Extraction and Analysis

Three biological replicates, each including eight apical or basal regions of the hypocotyl, were collected at T0, T1, T4, and T8. Metabolites were extracted from the frozen tissues and analyzed as previously described [68] with some modifications. Fresh plant material (0.1 g) was homogenized in liquid nitrogen and incubated in 1 mL of a cold (−20 °C) extraction mixture of methanol/water (80/20, volume/volume (*v*/*v*)) for 30 min at 4 °C. The solids were separated by centrifugation (20,000× *g*, 15 min at 4 °C) and re-extracted for another 30 min at 4 °C with 1 mL of extraction solution. The pooled supernatants were passed through Sep-Pak Plus C18 cartridges to remove interfering lipids and some plant pigments. The supernatants were collected and evaporated under vacuum at 40 °C. The residues were dissolved in 0.2 mL methanol/water (20/80, *v*/*v*) solution using an ultrasonic bath. The dissolved samples were filtered through Millex filters with a 13 mm diameter and 0.22 µm pore size nylon membranes (Millipore, Bedford, MA, USA) and placed into opaque microcentrifuge tubes.

Ten microliters of filtered extract was injected into a U-HPLC–MS system consisting of an Accela Series U-HPLC (Thermo Fisher Scientific) coupled to an Exactive mass spectrometer (Thermo Fisher Scientific) using a heated electrospray ionization interface. Mass spectra were obtained using Xcalibur software version 2.2 (Thermo Fisher Scientific). Metabolites of interest were identified by extracting the exact mass from the full-scan chromatogram obtained in negative mode and adjusting a mass tolerance of ≤1 ppm. The concentrations were semi-quantitatively determined from the extracted peaks using calibration curves of analogous compounds commercially available (OlChemIm, Olomouc, Czech Republic).

### 4.7. Statistical Analyses

The descriptive statistics were calculated by using the StatGraphics Centurion XV software (StatPoint Technologies Inc., Warrenton, VA, USA) and SPSS 21.0.0 (SPSS Inc., Chicago, IL, USA) programs. Outliers were identified and excluded from posterior analyses as described elsewhere [15]. We performed multiple testing analyses using the ANOVA *f*-test or Fisher’s least significant difference methods (*p*-value < 0.01, unless otherwise indicated). Nonparametric tests were used when necessary (i.e., AR emergence and AR number.

## 5. Conclusions

Taken together, our results indicate that wounding induced broad metabolic reprogramming (photosynthesis reactivation, photorespiration induction, and increased glycolysis) of some cells in the apical region of hypocotyls, providing the energy and structural elements required for rapid proliferation during initial callus growth. These phenomena are essential for the fate reprogramming that is required for de novo shoot formation.

## Figures and Tables

**Figure 1 ijms-22-10112-f001:**
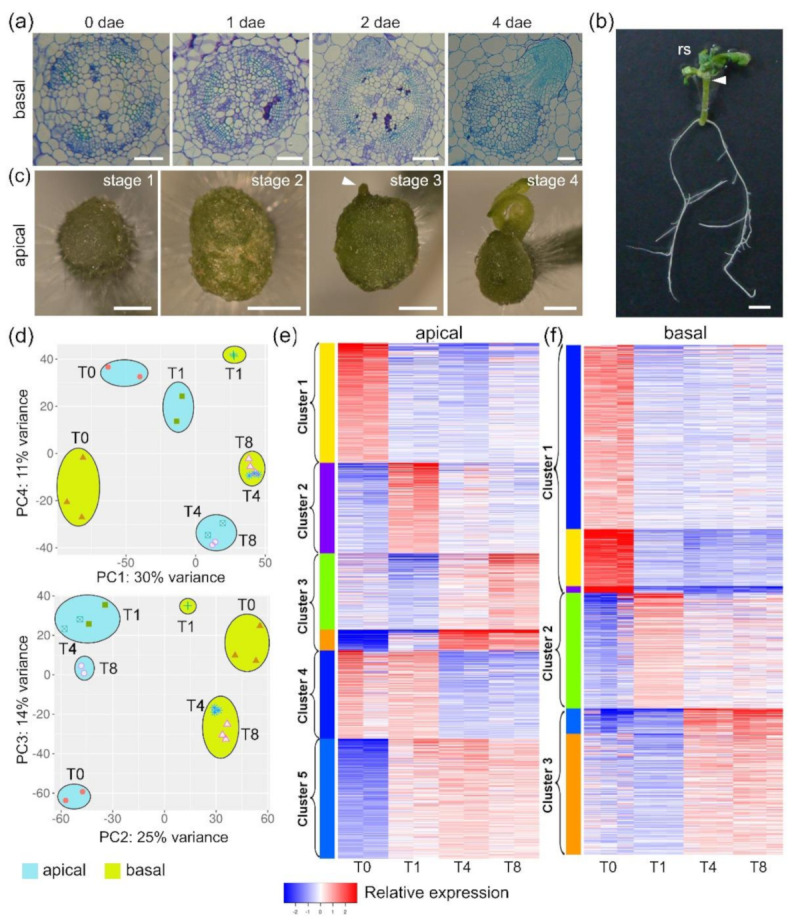
Wound-induced organ formation in tomato hypocotyl explants. (**a**) Adventitious root (AR) development in the basal region of hypocotyl explants; (**b**) whole-plant regeneration from hypocotyl and stem node explants at 21 dae; rs: regenerated shoot; (**c**) developmental stages during de novo shoot formation in the apical region of hypocotyls. Scale bars: 100 µm (**a**), 1 mm (**b**), and 4 mm (**c**); (**d**) principal component (PC) analysis of the RNA-Seq results; (**e**,**f**) k-means clustering of the time-course RNA-seq results from the apical (**e**) and basal (**f**) regions of hypocotyls. Predicted clusters are colored and grouped according to their temporal expression profile. Expression values are relative and adjusted to −2 and +2 values.

**Figure 2 ijms-22-10112-f002:**
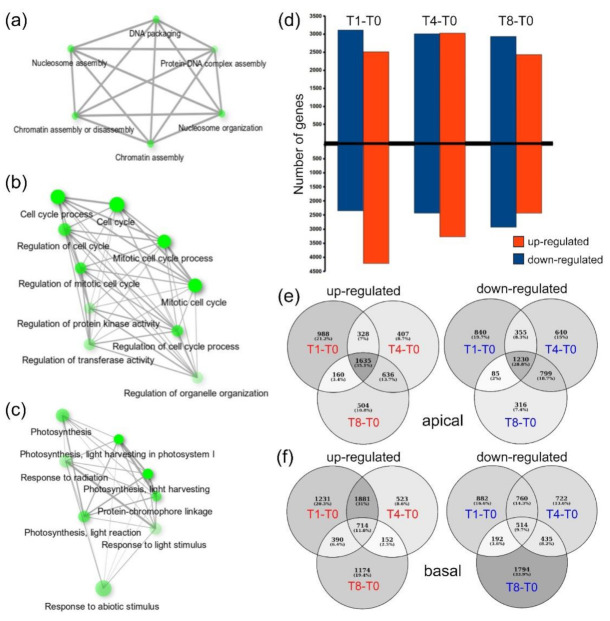
Functional enrichment and DEG analysis of wound-induced regeneration. (**a**–**c**) GO term-enriched networks of the upregulated genes in the apical region in the T1–T0 contrast (**a**), common upregulated genes in the T4–T0 and T8–T0 contrasts between the apical and basal regions (**b**), and specific upregulated genes in the apical region in the T4–T0 and T8–T0 contrasts (**c**); (**d**) number of DEGs found in each contrast, including the genes specific to the apical and basal regions; (**e**,**f**) Venn diagrams of upregulated and downregulated DEGs from the apical (**e**) and basal (**f**) regions of hypocotyls.

**Figure 3 ijms-22-10112-f003:**
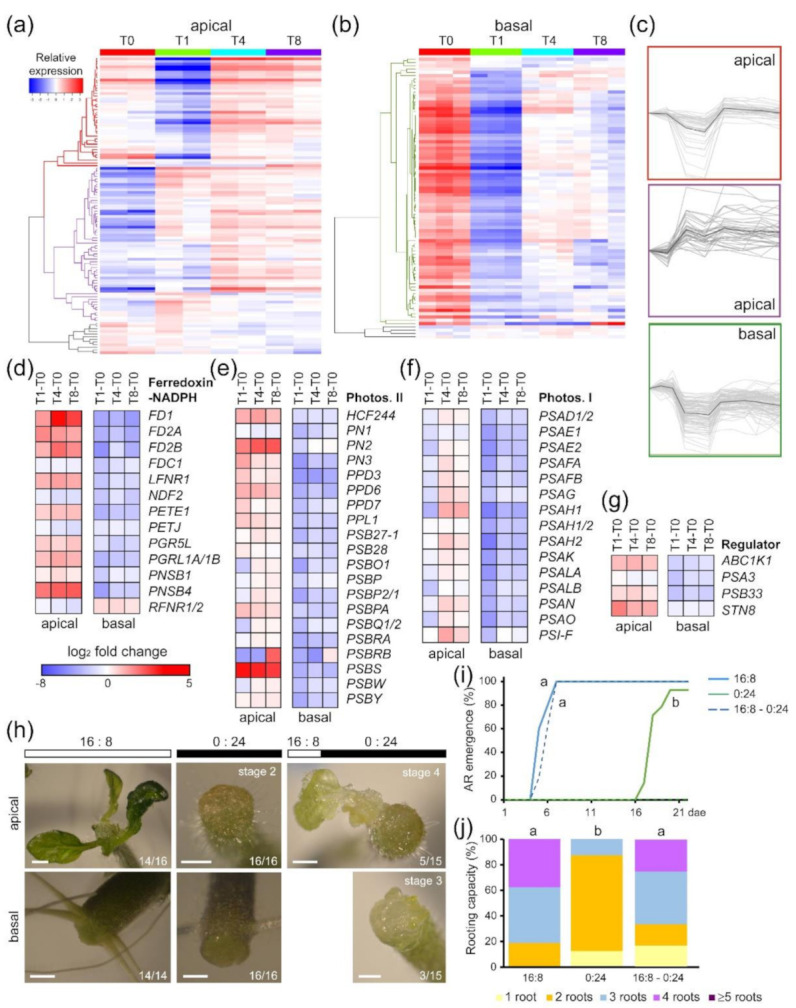
Regulation of photosynthesis-related genes during wound-induced organ formation. (**a**,**b**) Hierarchical clustering of the expression of photosynthesis-related genes in the apical (**a**) and basal (**b**) regions of hypocotyl explants over time; (**c**) expression profiles of some of the identified gene clusters; (**d**–**g**) DEGs involved in known photosynthesis subcomplexes, such as (**d**) ferredoxin: NAPDH reductase, (**e**) photosystem II, or (**f**) photosystem I, and in (**g**) key regulators; (**h**) representative images of wound-induced organ formation in the apical and basal regions of hypocotyl explants under different photoperiod conditions: 21 day standard photoperiod (16 h of light and 8 h of darkness; 16:8), 21 day darkness (0 h of light and 24 h of darkness; 0:24), and 3 day 16:8 photoperiod followed by 18 day 0:24 photoperiod (see Section 4); (**i**) adventitious root (AR) emergence; (**j**) rooting capacity of hypocotyl explants at 14 dae. Different letters indicate significant differences (*p*-value < 0.01) between the assay conditions. Gene annotations (**d**–**g**) are found in Appendix A. Scale bars: 1 mm. Expression values are log_2_ fold change.

**Figure 4 ijms-22-10112-f004:**
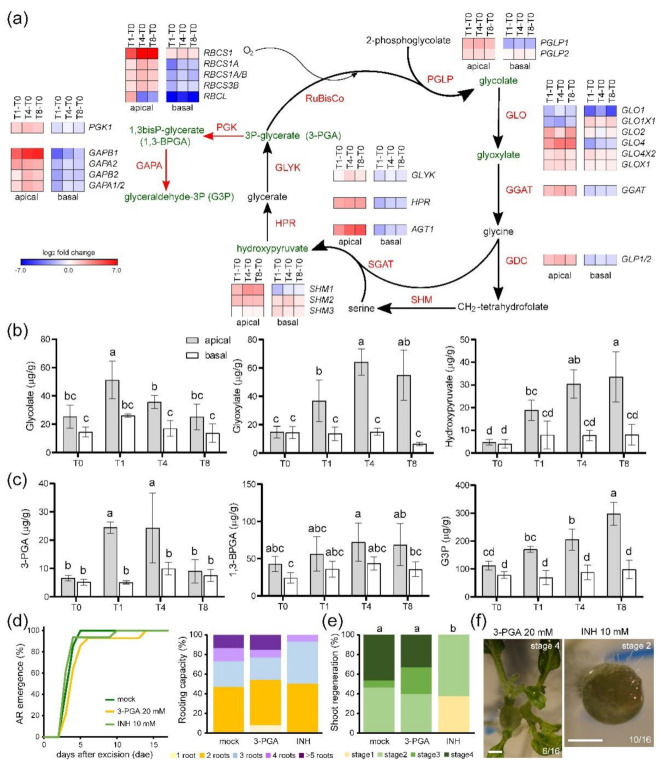
Photorespiration pathway activation during wound-induced organ formation. (**a**) Schematic photorespiration pathway representation. DEGs involved in photorespiration. Genes encoding enzymes of different steps in the pathway are shown: PGLP, phosphoglycolate phosphatase; GLO, glycolate oxidase; GGAT, glutamate-glyoxylate aminotransferase; GDC, glycine decarboxylase complex; SHM, serine hydroxymethyltransferase; SGAT, serine-glyoxylate aminotransferase; HPR, hydroxypyruvate reductase; GLYK, glycerate kinase. Two enzymes of the Calvin–Benson cycle are also included: PGK, phosphoglycerate kinase and GAPA, glyceraldehyde-phosphate dehydrogenase; (**b**,**c**) endogenous levels of several metabolites of the photorespiration pathway in the apical and basal regions of hypocotyl explants throughout the studied time course. Metabolites are indicated in green in panel (**a**); (**d**) AR emergence; (**e**) rooting capacity of hypocotyl explants at 14 dae; (**f**) representative images of de novo organ formation in the apical region of hypocotyl explants grown on SGM that were supplemented with either 20 mM 3-PGCA or 10 mM isoniazide (INH) in their apical region. Different letters indicate significant differences (*p*-value < 0.01) between the assay conditions. Gene annotations in panel (**a**) are found in Appendix A. Scale bars: 1 mm. Expression values are log_2_ fold change.

**Figure 5 ijms-22-10112-f005:**
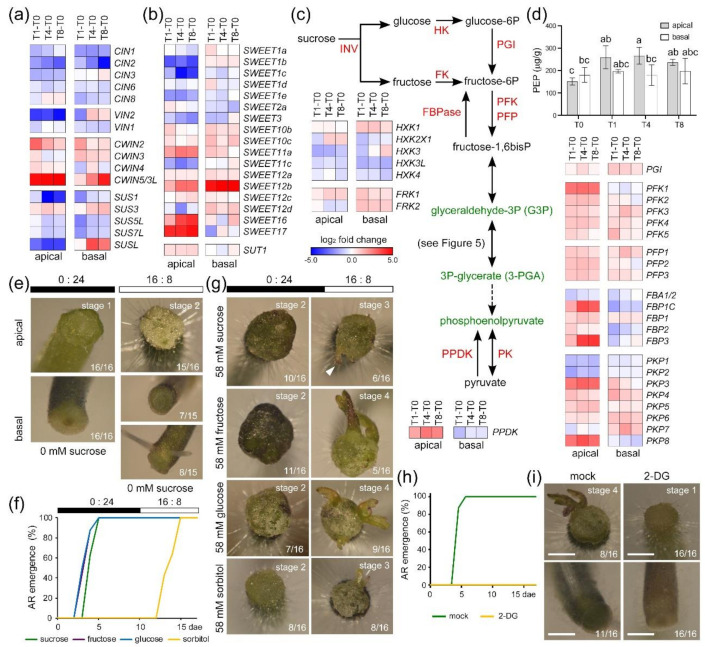
Sucrose is required for wound-induced organ formation. (**a**) DEGs encoding invertases (CIN, VIN, and CWIN) and sucrose synthases (SUS); (**b**) DEGs encoding putative sugar transporters of the SWEET and SUT families; (**c**) schematic of the glycolysis and gluconeogenesis pathways and DEGs encoding enzymes in these pathways. Genes encoding key enzymes of different steps in the pathway are shown: INV, invertase; HK, hexokinase; FK, fructokinase; PGI, phosphoglucose isomerase; PFK/PFP, phosphofructokinase; FBPase, fructose bisphosphatase; PK, pyruvate kinase; PPDK, pyruvate-phosphate dikinase; (**d**) endogenous levels of phosphoenolpyruvate in the apical and basal regions of hypocotyl explants over time; (**e**) representative images of de novo organ formation in the apical and basal regions of hypocotyl explants under different photoperiod conditions (16:8 and 0:24) without sucrose; (**f**) AR emergence of hypocotyl explants grown for 10 days under a 0:24 photoperiod followed by 7 days under a 16:8 photoperiod; (**g**) representative images of shoot formation stages in the apical region of hypocotyl explants grown in the presence of different sugars under the same photoperiod conditions as in panel (**f**); (**h**) AR emergence of hypocotyl explants grown for 14 dae on SGM (mock and 2-DG) under a 16:8 photoperiod. Different letters in panels (**d**,**f**,**h**) indicate significant differences (*p*-value < 0.01) between the assay conditions; (**i**) representative images of de novo organ formation in the apical and basal regions of hypocotyl explants grown on SGM (mock and 2-DG) under a 16:8 photoperiod. Gene annotations in panels (**a**–**c**) are found in Appendix A, respectively. Scale bars (**e**,**g**,**i**): 1 mm.

## Data Availability

Raw sequence files and read count files are publicly available in the NCBI’s BioProject repository upon publication (PRJNA731333). Gene functional annotation is available in the Appendix A of this article. All other data that support the findings of this study are available from the corresponding author upon reasonable request.

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
