# Peer review of "Tissue-Specific Metabolic Reprogramming during Wound-Induced Organ Formation in Tomato Hypocotyl Explants"

_ijms, 2021, doi:10.3390/ijms221810112_

Round 1
Reviewer 1 Report
In this manuscript authors studied the tissue-specific gene and metabolite reprogramming during wound-induced organ formation in tomato hypocotyl explants. It is an interesting study adding lot value to developmental biology and deserves publication in IJMS. Before accepting the manuscript for publication, the following comments need to be addressed
- All DEG list with the fold change values should be given as supplemental table.
- There is no clarity on how many biological replicates have been sequenced.
Reviewer 2 Report
The manuscript “Tissue-specific metabolic reprogramming during wound-induced organ formation in tomato hypocotyl explants” reported the role of metabolism in wound-induced organ formation under in vitro conditions. In this study, hypocotyl explants obtained from tomato ‘Micro-Tom’ were used for transcriptome and metabolomics analyses to identify regulatory mechanisms that participate in adventitious-shoot and -root regeneration. The results showed that exogenous sucrose was essential for effective shoot regeneration in the apical region of the hypocotyl explants but was not necessary for root formation when another energy source was present (e.g., photosynthesis/photorespiration-derived compounds). The manuscript is well-written and needs minor corrections.
Some of the specific comments:
L44-46: Please add more information on the role of sugars, amino acids, and polyamines. Sucrose is a frequent carbon source in tissue culture media that plays an important role in culture initiation and development and metabolite production. Citation required.
L54: tomato (Solanum lycopersicum L.) ‘Micro-Tom’ instead of ‘Micro-Tom’ cultivar of tomato (Solanum lycopersicum L.). Please delete cultivar throughout the text.
L85: days after excision (dae) instead of dae; Benefits of enhanced terminal room (BETR) instead of BETR. Please, before use of abbreviations, give the full term at first use.
L148: Please check the DEGs numbers.
L162: Please correct the figure legend according to Figure 2.
L167: Please check the DEGs numbers.
L168: (Figure 2f).?
L170 and 198: ‘photosynthesises’ in general, most in vitro explant cultures are unable to photosynthesize efficiently due to a lack of chlorophyll in explants and the presence of sugar.
L227: Please indicate the different light conditions in Figure 3h.
L254: Please change the abbreviation: GO, glycolate oxidase to avoid the confusion ‘Gene Ontology (GO)’
L287: Please provide supporting UHPLC chromatograms of glycolate, glyoxylate, and hydroxypyruvate.
L396: Please indicate the compounds.
L489: Please indicate the age of hypocotyl explants.
L494: Please indicate the culture conditions.
L579: The authors fail to make conclusions.
